# Effect of Edible Carboxymethyl Chitosan-Gelatin Based Coating on the Quality and Nutritional Properties of Different Sweet Cherry Cultivars during Postharvest Storage

Yu-Lei Zhang [1], Qing-Liang Cui [1,*], Yu Wang [2], Fei Shi [2], Hua Fan [3], Yan-Qing Zhang [1], Si-Tong Lai [1], Ze-Hui Li [1], Lang Li [1] and Yi-Ke Sun [1]

1    College of Agricultural Engineering, Shanxi Agricultural University, Taigu 030801, China; zyl0101@stu.sxau.edu.cn (Y.-L.Z.); yqzhang@sxau.edu.cn (Y.-Q.Z.); lsitong951116@stu.sxau.edu.cn (S.-T.L.); Z20193273@stu.sxau.edu.cn (Z.-H.L.); langli@stu.sxau.edu.cn (L.L.); yikesun@stu.sxau.edu.cn (Y.-K.S.)
2    College of Food Science and Engineering, Shanxi Agricultural University, Taigu 030801, China; wangyu@sxau.edu.cn (Y.W.); shifei@sxau.edu.cn (F.S.)
3    Experimental Teaching Center, Shanxi Agricultural University, Taigu 030801, China; fanhua@sxau.edu.cn
*    Correspondence: qlcui@sxau.edu.cn; Tel.: +86-0354-6288906

**Abstract:** Sweet cherry has many cultivars with different storability and nutritional properties. To reveal the reasons for the differences in storability among cultivars and improve the quality of sweet cherries, the surface microstructure of four representative sweet cherry cultivars (Red Light, Ranier, Red Agate, Friendship) epidermis and peduncle at harvest were examined and the effects of carboxymethyl chitosan-gelatin (CMCS-GL) based edible coating incorporating $CaCl_2$ and ascorbic acid (AA) (AA-$CaCl_2$-CMCS-GL) on the quality and nutritional characteristics of sweet cherry were evaluated. Results showed there were significant differences in the wax distribution of the epidermis and the number of stomata on the peduncle surface between four cultivars of sweet cherries at harvest, which was closely related to fruit decay ratio during storage. AA-$CaCl_2$-CMCS-GL coating delayed the onset of decay and the fruit decay ratio in coated groups (3.0%–15.3%) was significantly lower than in control groups (17.7%–63.0%) after 33 d storage. The coating also helped to maintain the quality and nutritional characteristics of four sweet cherry cultivars, including reducing weight loss, maintaining better skin color, peduncle freshness, higher fruit firmness, titratable acidity, AA, total phenolics content, total anthocyanins concentration, and antioxidant capacity. These results suggested that AA-$CaCl_2$-CMCS-GL coating could be considered as a new preservation method for improving postharvest quality and nutritional properties of different sweet cherry cultivars.

**Keywords:** carboxymethyl chitosan; gelatin; sweet cherry; cultivar; nutritional properties; peduncle freshness

## 1. Introduction

Sweet cherries (*Prunus avium L.*) are valued by consumers because of their precocity, bright color, sweet and sour taste, rich nutrition, and antioxidant activity [1,2]. Production and consumption of sweet cherries have increased in recent years as consumers have appreciated their wealth of polyphenols and other substances beneficial to human health [1–3]. Sweet cherries are mostly consumed fresh [1] and are highly perishable [4].

Edible coatings are made from natural food-grade materials such as polysaccharides, proteins, lipids, or their complexes, which can be eaten. Chitosan (CS)-based edible coating has been used to improve the quality and shelf life of sweet cherries during storage [5–8], However, chitosan is insoluble in water and general organic solvents, it can only be dissolved in acidic solutions below pH 6.3, which is not convenient for consumers to clean before eating, and the bad smell produced by acidic solutions may affect the aroma of fruits and vegetables, which limits its application in the preservation of fruits and vegetables [9]. Carboxymethyl chitosan (CMCS) is an amphoteric derivative of CS. Due to its large number

of carboxyl groups, it has better water solubility, biocompatibility, and biodegradability than CS [10,11]. CMCS can be dissolved at a wider pH range, including pure water, and, because of its good moisture and oxygen barrier properties, it can be used as a food packaging film [12]. Pure CMCS film is brittle, strongly hydrophilic, and has poor barrier and mechanical properties which limits its application in food packaging [12,13]. Gelatin (GL) is a natural water-soluble protein obtained by partially hydrolyzing collagen. Studies have shown that blending CS and GL improves the properties of edible films [14–16], but there is no research on the application of CMCS-GL based edible coating in the preservation of fruits and vegetables.

Edible coatings can be used as carriers of antioxidants, antibacterial agents, anti-browning agents, and nutritional fortifiers [17–19]. $CaCl_2$ can be added to CS-based film as a crosslinking agent to improve the performance of edible film, and calcium ions can form an ionic bridge with the residues of the free carboxyl group of pectin galacturonic acid to strengthen the structure of the cell wall, thus improving the quality of fruit during storage [20]. As a safe and inexpensive antioxidant with a good antioxidant effect, ascorbic acid (AA) is an essential vitamin for human health. It can be obtained from fruits and vegetables, and AA has been proved to be effective in controlling the enzymatic browning of fruits and vegetables [21,22]. Studies also have shown that the mixing of CS with $CaCl_2$ or/and AA improved the preservation of strawberries [23,24], pear [25,26], plum [21], papaya [27], litchi [28], fresh-cut honeydew melon [29], fresh-cut apples [22]. Therefore, in this experiment, the CMCS-GL based edible coating incorporating $CaCl_2$ and AA (AA-$CaCl_2$-CMCS-GL) was prepared for the preservation of sweet cherries. Many studies on the preservation of sweet cherries using edible coatings have focused on a single cultivar [5,7,30–33] with only a few studies investigating two or three cultivars [34,35]. However, different cultivars of sweet cherries have obvious differences in storability and nutritional quality. Therefore, it is necessary to select several representative cultivars for the coating experiment to examine the applicability of an edible coating to different cultivars of sweet cherries.

There are hundreds of sweet cherry cultivars, different genetic determines their differences in harvest time, peel color, fruit size, taste, and storability, etc. [2,36]. Among them, storability is a key factor that determines its commercial value. The epidermis tissue structure of fruits is closely related to its storability. More and more studies have shown that the epidermis wax of the fruit plays an important role in the quality, storability, and pathogen sensitivity of the fruit after harvest. For example, the wax content of the fruit epidermis directly determines the rate of water loss after harvested [37,38]. The stomata on the fruit epidermis are the main channels for its transpiration, and its state and quantity are also closely related to the postharvest water loss of the fruit. Other studies have shown that the stomatal density in the peduncle surface of sweet cherry is 40 times higher than that of the fruit surface, the permeability of the peduncle surface is higher than that of the fruit surface, the wax on the surface of the peduncle is the main rate-limiting barrier for its water loss by transpiration [39], which indicates that although the peduncle is not edible, it plays an important role in the storability during postharvest storage as a part of sweet cherries. However, there is no research on the wax and stomata distribution on the surface of the epidermis and peduncle for different sweet cherry cultivars.

Thus, the present study was to investigate the surface microstructure of the epidermis and peduncle for four sweet cherry cultivars (Red Light, Ranier, Red Agate, Friendship) and the effectiveness of AA-$CaCl_2$-CMCS-GL edible coating on preserving the quality (including the fruit quality and peduncle freshness) and nutritional properties of four sweet cherry cultivars during postharvest storage.

## 2. Materials and Methods

### 2.1. Materials

2.1.1. Plant Materials

The four cultivars of sweet cherry with different types of maturity were harvested from 17 years old trees owned by the Pomology Institute, Shanxi Agricultural University (Taigu, Shanxi Province, China; 112°50′ E, 37°34′ N) during 2020 (Figure S1). The cultivar (abbreviation), types of maturity, harvest date, skin color, pulp color, and average fruit weight of sweet cherry are presented in Table 1.

**Table 1.** The cultivar (abbreviation), types of maturity, harvest date, skin color, pulp color, and average fruit weight of sweet cherry

| Cultivar (Abbreviation) | Types of Maturity | Harvest Date | Skin Color | Pulp Color | Average Fruit Weight (g) |
|---|---|---|---|---|---|
| Red Light (RL) | Precocious | 27 May | Fuchsia | Red | $5.16 \pm 0.15$ |
| Ranier (RN) | Mid-ripening | 6 June | Yellow-red | Light yellow | $4.35 \pm 0.12$ |
| Red Agate (RA) | Mid-ripening | 6 June | Fuchsia | Fuchsia | $5.98 \pm 0.14$ |
| Friendship (FS) | Serotinous | 15 June | Crimson | Red | $7.86 \pm 0.19$ |

After randomly sampling at the commercial ripening stage, fruit were immediately transported to cold storage ($0 \pm 0.5$ °C, RH 85%–90%) in the laboratory until their core temperature dropped to 0 °C. Fruit were then screened for color, size uniformity, mechanical damage, disease, and insect pests. Finally, each cultivar of cherries was randomly divided into two groups.

2.1.2. Chemicals

CMCS, GL, glycerol, $CaCl_2$, L-AA, 2,2-Diphenyl-1-picrylhydrazyl(DPPH), and glutaraldehyde, all analytical pure, were purchased from Macklin Biotechnology Co. Ltd. (Shanghai, China). Anhydrous ethanol, tween-20, potassium persulfate, calcium carbonate powder, acetone, quartz sand, all analytical pure, purchased from Tianjin Damao Chemical Reagent Factory, Tianjin, China; Methanol, potassium chloride, all analytical pure, purchased from Tianjin Fengchuan Chemical Reagent Technology Co., Ltd., Tianjin, China, 2,6-dichlorophenol indophenol sodium, Folin-Ciocalteu reagent, all analytical pure, purchased from Beijing Solaibao Technology Co., Ltd., Beijing, China; Gallic acid standard product, AA standard product, Beijing Solai bao Technology Co., Ltd. The rest of the chemical reagents were analytical grade.

### 2.2. Experimental Design

2.2.1. Preparation of AA-$CaCl_2$-CMCS-GL Edible Coating Solution

Solutions of CMCS and GL at 2% (*w/v*) in distilled water were prepared separately by heating in a 60 °C water bath with continuous stirring until completely dissolved (~30 min), then cooled to $23 \pm 1$ °C. The solutions were mixed in a mass ratio of 2:1 (CMCS: GL, *w:w*), then based on the weight of the mixed solution of CMCS and GL, 2% $CaCl_2$ powder, and 2% L-AA were added to the solution as the crosslinking agent and antioxidant with stirring at $23 \pm 1$ °C for 30 min in the presence of 1% glycerol and 0.1% tween-20 acting as plasticizer and surfactant (improve the wettability of coating solution on the surface of sweet cherry), respectively. After stirring overnight at $23 \pm 1$ °C, the mixture was centrifuged for 10 min at $4000 \times g$ to remove air bubbles and particulates (the impurities and insoluble components), and the supernatant was collected for later use.

2.2.2. Sweet Cherry Treatments and Storage Conditions

The four cultivars of sweet cherries were washed in tap water, dried at $23 \pm 1$ °C, immersed in AA-$CaCl_2$-CMCS-GL edible coating solution for 2 min (including the fruit and its peduncle), then placed in a tray and dried completely in the air at $23 \pm 1$ °C. The

control group was washed with tap water only. Treated and control samples were packed into 0.2 mm polyethylene terephthalate plastic boxes ($157 \times 118 \times 72$ mm$^3$) with mesh holes, then stored at $0 \pm 0.5$ °C, RH 85%–90% for 30 d. Samples were taken every 6 d to measure the physical and chemical indicators. All determinations were carried out at $23 \pm 1$ °C, 2–3 h after removal from cold storage. After being refrigerated for 30 d, the remaining samples were stored at $23 \pm 1$ °C, RH 40%–50%, for 3 d to simulate shelf storage at room temperature, and samples were taken for analysis. As shown in Table 2, there were eight groups including the control.

**Table 2.** The four cultivars of sweet cherry with or without the treatment of AA-CaCl$_2$-CMCS-GL coating (2 min).

| Serial Number | Cultivar | Treatment | Abbreviation |
| :---: | :---: | :---: | :---: |
| 1 | RL | Tap water | RL-C |
| 2 | | AA-CaCl$_2$-CMCS-GL coating | RL-T |
| 3 | RN | Tap water | RN-C |
| 4 | | AA-CaCl$_2$-CMCS-GL coating | RN-T |
| 5 | RA | Tap water | RA-C |
| 6 | | AA-CaCl$_2$-CMCS-GL coating | RA-T |
| 7 | FS | Tap water | FS-C |
| 8 | | AA-CaCl$_2$-CMCS-GL coating | FS-T |

*2.3. The Surface Microstructure Observation of Sweet Cherry Epidermis and Peduncle*

A scalpel was used at harvest to sample the skin and peduncles of the four cherry cultivars, cutting the skin approximately 5 mm in length and 4 mm in width on the equatorial plane, and the middle of the peduncle approximately 5 mm in length. These were quickly placed in 3% glutaraldehyde fixative solution (0.1 mol·L$^{-1}$, pH 7.0 phosphate buffer solution) and fixed at 4 °C for 2 d. After fixation, samples were rinsed three times with pH 7.0 (0.1 mol·L$^{-1}$) phosphate buffer solution for 15 min then dried progressively in alcohol. Step by step dehydration used ethanol at concentrations of 30%, 50%, 70%, 80%, 90%, and 95%, for 15 min each time, followed by 100% ethanol dehydration twice for 20 min, then replacing the solvent with acetone. After critical point drying (model K850; Quorum Technologies Ltd., East Sussex, UK) was completed, the dried material was stuck on a sample stage with conductive tape, sprayed with platinum using an ion sputtering coater (model JFC-1600; JEOL, Tokyo, Japan) then placed under a scanning electron microscope (model JEM-6490 LV; JEOL, Tokyo, Japan) for morphological examination.

*2.4. Measurement of Fruit Decay Ratio, Weight Loss, and Fruit Firmness*

The fruit decay ratio of sweet cherry was determined by the methods of Wang et al. [40] and Aglar et al. [41]. One hundred randomly selected fruit per replicate were counted and three replicate analyses were carried out.

The decay ratio was evaluated visually and calculated as follows:

$$\text{Fruit decay ratio (\%)} = (A_1 / A_2) \times 100 \tag{1}$$

where $A_1$ is the number of rotten fruit and $A_2$ the total number of fruit. Rotten fruit were classified as having at least one visible rotten spot on the surface.

Weight loss of sweet cherry was expressed as a percentage of the initial total sample weight. A digital scale ($\pm 0.01$ g; MAX-C6002; Shenzhen, China) was used to weigh approximately 300 g of fruit from each group, and determinations were performed in triplicate.

A texture analyzer (TA-TX Plus; Stable Microsystems, Godalming, UK), equipped with a cylindrical probe (P/36R), was used to measure fruit firmness. The compression speed was 2 mm/s and the compression depth 5 mm. The maximum force that the fruit endure during the compression process was defined as fruit firmness, and the result was expressed in N. Ten fruit from each group were tested and the mean value was calculated.

### 2.5. Measurement of Skin Color Characteristics

A colorimeter (model CM–5; Konica-Minolta, Tokyo, Japan) was used to measure the skin color ($L^*$, $a^*$, and $b^*$) of sweet cherry on opposite sides of 30 fruit. The $D_{65}$ was used as illuminant, and the observation angle was $10°$. $L^*$, chroma, and hue angle values were used to describe the skin color characteristics using the following calculations [6,21,41]:

$$\text{Chroma} = (a^{*2} + b^{*2})^{1/2} \tag{2}$$

$$\text{hue angle} = \tan^{-1}(b^*/a^*) \tag{3}$$

### 2.6. Measurement of Peduncle Browning Incidence (PBI), Peduncle Moisture Content (PMC), and Peduncle Chlorophyll Content (PCC)

The PBI and PMC of sweet cherry were determined by the methods of Wang et al. [40] and Aglar et al. [41]. One hundred randomly selected fruit were counted, and three replicate analyses were carried out.

The PBI was calculated as follows:

$$\text{PBI (\%)} = (A_1/A_2) \times 100 \tag{4}$$

where $A_1$ is the number of fruit with >30% stem surface browned during postharvest storage and $A_2$ is the total number of fruit.

The PMC was calculated as follows:

$$\text{PMC (\%)} = [(A_1 - A_2)/A_1] \times 100 \tag{5}$$

where $A_1$ is the original weight of the fresh peduncle and $A_2$ is the final peduncle weight after drying at 80 °C.

PCC was determined with the acetone extraction method of Mackinney [42] with some modification. Chlorophyll in the peduncle was extracted with 80% acetone and the absorbance of the extract measured at 645 nm and 663 nm. PCC was calculated as follows:

$$\text{PCC (mg·kg}^{-1}) = 20.29\ A_{645} + 8.05\ A_{663} \tag{6}$$

### 2.7. Measurement of Soluble Solids Content (SSC), Titratable Acidity (TA), and AA Content

For the measurement of SSC and TA, 30 fruit were randomly selected from each group and weighed after removing the cores. A juicer (model WBL2501B; Midea, Guangdong, China) was used to extract the juice at 2000 g. The juice was centrifuged for 20 min at $10,000\times g$ at 4 °C (model 5804R; Eppendorf, Germany). SSC in the supernatant was measured using a Pocket Refractometer (model PAL-1; Atago, Tokyo, Japan). TA was measured by titrating 20 mL of diluted fruit juice with 0.1 M NaOH solution to pH 8.1. SSC and TA were expressed as % and g malic acid per kg fresh weight (g·kg$^{-1}$), respectively [7,41].

The AA content of sweet cherry was determined using the 2,6-dichloroindophenol titrimetric method of Zam [8] with a slight modification. To avoid interference from the AA added to the edible film, AA content was only determined in the cherry pulp. Fruit pulp (~10 g) mixed with 20 g/L oxalic acid solution was ground in an ice-cooled mortar. The extract was centrifuged at 4 °C for 20 min at $10,000\times g$ and the supernatant titrated with 2,6-dichloroindophenol to a rose pink color lasting for 15–20 s. AA concentration was expressed as mg per kg fresh weight (mg·kg$^{-1}$).

### 2.8. Measurement of Total Phenolics Content (TPC), Total Anthocyanins Concentration (TAC), and Antioxidant Activity

Preparation of extraction solution: fruit tissue (2.0 g) was homogenized with a little precooled 1% HCl-methanol solution in an ice bath, then transferred to a 20 mL graduated, stoppered test tube, and made up to 20 mL with 1% HCl-methanol solution. After extracting for 20 min at 4 °C, the solution was centrifuged at 4 °C for 20 min at $10,000\times g$. The

supernatant was collected and used for the determination of total phenolic and anthocyanin concentrations, and antioxidant activity.

Folin–Ciocalteu reagent was used to determine TPC based on the method described by Abdipour et al. [7]. The absorbance at 725 nm of the solution after the reaction was measured by a UV-visible spectrophotometer (P4; Shanghai Mapada Instruments, Shanghai, China). TPC was expressed as mg gallic acid equivalent (GAE) per g fresh weight (mg·kg$^{-1}$) (Figure S2).

TAC was determined by the pH differential method described by Aglar et al. [41] and expressed as mg of cyanidin-3-glucoside equivalents (cy-3-glu) per fresh weight (mg·kg$^{-1}$).

Antioxidant activity was evaluated based on 2,2-diphenyl-1-picrylhidrazyl (DPPH) radical scavenging capacity (RSC) method proposed by Abdipour et al. [7] with some modification. A standard curve was plotted using different concentrations of AA (0–100 µg·mL$^{-1}$) (Figure S3). The results were expressed as mg ascorbic acid equivalent (AAE) per g FW (g·kg$^{-1}$). All analyses were performed in triplicate.

### 2.9. Statistical Analysis

The experiment was performed using a randomized design. Results were expressed as mean ± standard deviation and analyzed by one-way analysis of variance using IBM SPSS Statistics for Windows (SPSS Inc., Chicago, IL, USA). Duncan's multiple range test was used for mean separations and $p < 0.05$ was considered significant. Relationships between different parameters were determined by Pearson's correlation.

### 3. Results

### 3.1. Microstructure of Sweet Cherry Epidermis and Peduncle Surface at Harvest

Differences in stomatal morphology, wax distribution, and morphology around the stomata are shown in Figure 1. Stomata on the peel of RL were small with a small amount of granular wax and more filamentous wax scattered around (Figure 1a); stomata on RN were slightly larger with a small amount of granular and filamentous wax around the stomata (Figure 1b); stomata on RA were larger than on RL and RN, with more flaky wax distributed inside the stomata and a large amount of film-like wax distributed around the stomata (Figure 1c); while stomata on FS were larger than the other varieties with a large amount of granular wax distributed in and around the stomata and some of the granular wax connected by filamentous wax (Figure 1d).

Figure 1 also shows the distribution and morphology of epidermal wax on the skin. Most of the RL epidermal wax was filamentous with a small amount of granular wax (Figure 1e); a small amount of uneven granular wax connected by a small amount of filamentous wax was distributed on the epidermis of RN (Figure 1f); RA had more wax than RN on the epidermis and was granular with uneven sizes (Figure 1g); while FS had the most wax, covering almost all of the epidermis with large, waxy particles (Figure 1h).

Figure 2 illustrates the number of stomata, wax distribution, and morphology of the surface of the peduncles at ×300 magnification. There were ten stomata on the surface of the RL peduncle (the most abundant among the four varieties) with a small amount of granular wax distributed on its surface (Figure 2a); there were six stomata on the surface of the RN peduncle with a very small amount of wax (Figure 2b); RA had five stomata and more wax than RL and RN (Figure 2c); while the surface of the FS peduncle had no stomata visible to the naked eye at this magnification and the epidermis was covered with a large number of granular waxes of varying sizes (Figure 2d).

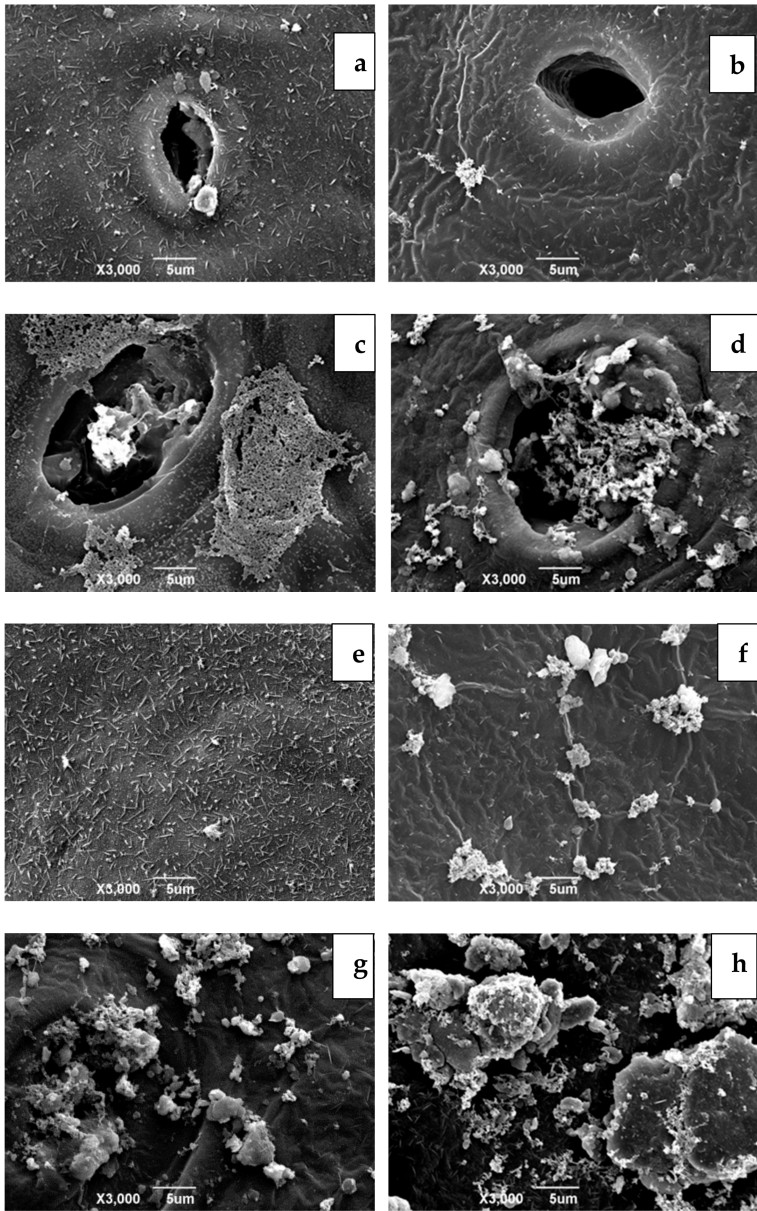

**Figure 1.** SEM micrographs of sweet cherry epidermis: (**a**) RL, (**b**) RN, (**c**) RA, (**d**) FS show the stomata, wax distribution, and morphology on the epidermis at ×3000 magnification; (**e**) RL, (**f**) RN, (**g**) RA, (**h**) FS show the wax distribution and morphology in areas other than the stomata at ×3000 magnification.

Figure 2 shows the stomata, wax distribution, and morphology of the peduncle epidermis at ×1200 magnification. A small amount of uneven granular wax was distributed on the surface of the RL peduncle with more wax distributed on the edge of the lower stomata (Figure 2e); there was a very small amount of granular wax on the surface of the RN peduncle (Figure 2f); more granular and filamentous wax was evident on the surface of RA peduncle (Figure 2g), while the surface of the FS peduncle showed a large amount of granular and filamentous wax and more wax around the stomata (Figure 2h).

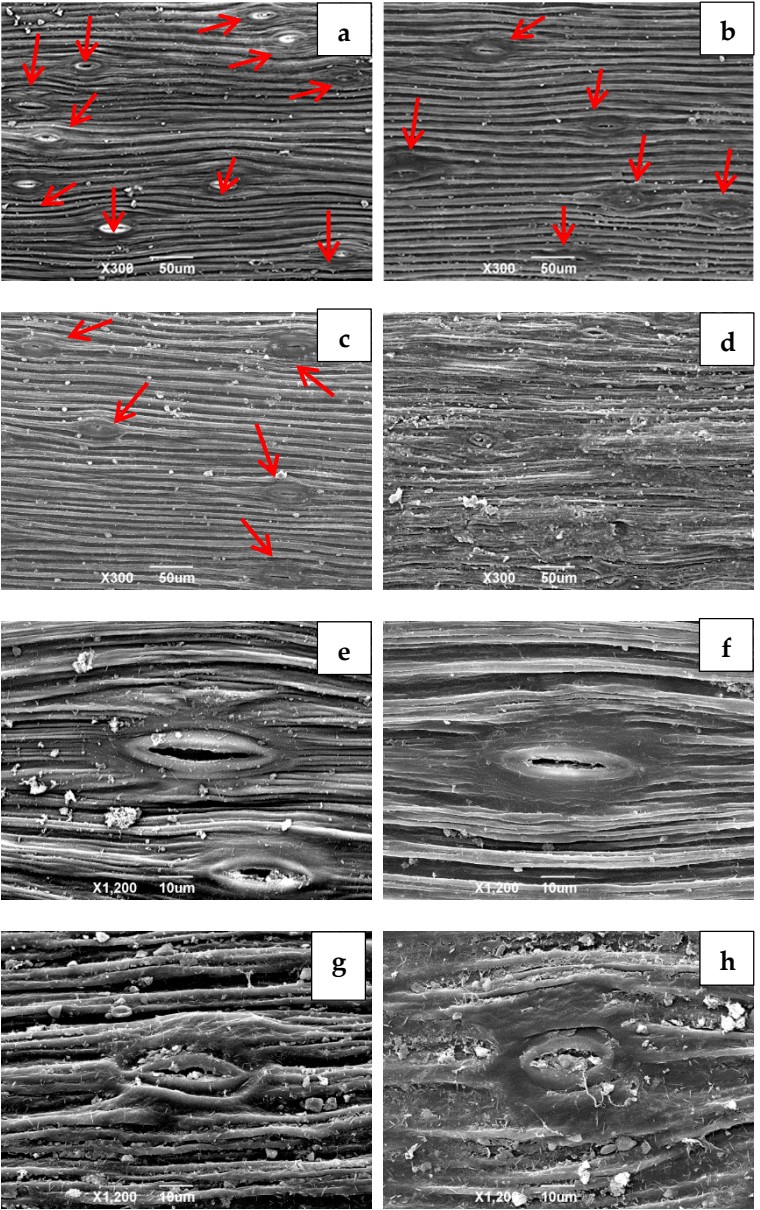

**Figure 2.** SEM micrographs of sweet cherry peduncle surface: (**a**) RL, (**b**) RN, (**c**) RA, (**d**) FS show the stomata number, wax distribution, and morphology on the peduncle at ×300 magnification; (**e**) RL, (**f**) RN, (**g**) RA, (**h**) FS show the stomata, wax distribution, and morphology at ×1200 magnification.

*3.2. Fruit Decay Ratio, Weight Loss, and Fruit Firmness*

The fruit decay ratio, weight loss, and fruit firmness during storage of control and treated sweet cherries are shown in Table 3. The time and rate of decay of the varieties varied during storage. RL-C was beginning to rot after 12 days of storage while FS-T did not show any rot until day 30. The fruit decay ratio of RL-C was significantly higher than the other varieties from day 12 to 30 and the ratios of RL-C, RN-C, and RA-C were significantly higher than FS-C on day 33. FS-T had the lowest decay ratio (3.0%) which showed that of the four varieties RL was the least suitable for long-term storage, followed by RN and RA, while FS could be used for long-term storage. The AA-CaCl$_2$-CMCS-GL coating treatment delayed the onset of decay and the decay ratio after treatment was significantly lower than the control groups.

Weight loss in all groups continued during storage (Table 3). From day 18 to day 33 of storage the weight loss of FS-C was the highest, followed by RN-C, both being significantly higher than RL-C and RA-C. There was no significant difference between RL-C and RA-C

on day 18 and 24. Of the control groups, weight loss in RA-C was the lowest by day 30 and 33 (8.01% and 10.75%, respectively) which were significantly lower than the other controls. From day 12 the weight loss of all the treatment groups was significantly lower than their control groups with RA-T consistently being the lowest. On day 33, the order of weight loss in the treatment group was RA-T < RL-T < RN-T < FS-T.

The fruit firmness of the cultivars was different at the beginning of storage. FS-C was the firmest (23.87 N), RA-C was the next firmest (20.73 N), while there was no significant difference between RL-C (19.49 N) and RN-C (19.65 N). During storage, the firmness in all groups decreased continuously. From day 12 firmness of the coated groups was significantly higher than the control groups. At the end of storage, the order of fruit firmness in the coated groups was RL-T < RN-T < FS-T < RA-T.

### 3.3. Skin Color Characteristics

The skin color of sweet cherry is used as an indicator of quality and ripeness, and thus determines acceptability by consumers [43]. At the beginning of storage, the four cultivars exhibited significant differences in skin color characteristics ($L^*$, chroma, and hue angle; Table 4). Values of $L^*$, chroma, and hue angle for RN-C were significantly higher than the other control groups. The value of $L^*$ in the four coated groups was significantly higher than the control groups.

During storage, the skin color in all groups darkened, with $L^*$ values decreasing, although this change was slower in coated groups than control groups. Except for RA, the chroma value tended to increase then decrease (RA showed a continuous downward trend). At the end of storage chroma and hue angle of all coated groups were significantly higher than their control groups.

### 3.4. PBI, PMC, and PCC

Green and turgid peduncles on sweet cherry are indicative of postharvest freshness, whereas yellow or brown and shriveled peduncles imply an aged fruit (Knoche et al., 2015; Linke et al., 2010). PBI, PMC, and PCC during storage of control and treated sweet cherries are shown in Table 5.

The PBI of all groups was zero at harvest but varied during storage. In the first 18 days of storage, PBI increased faster than from day 24 to day 33. From day 6 to the end of storage PBI in the coated groups was significantly lower than in their control groups, which indicated that the AA-CaCl$_2$-CMCS-GL coating significantly improved PBI. The PBI of RL-C was consistently the highest and FS-T was the lowest. At the end of storage, the order of PBI in treated and control groups was FS-T(FS-C) < RN-T(RN-C) < RA-T(RA-C) < RL-T(RL-C).

The PMC of RN and RA was significantly higher than RL and FS at harvest. During cold storage up to day 30, PMC of all groups showed a slow downward trend. During storage at room temperature (day 30 to 33), PMC in all control groups decreased rapidly. Compared with day 30, the PMC of RL-C, RN-C, RA-C, and FS-C decreased by 31.6%, 26.1%, 31.3%, and 14.2%, while RL-T, RN-T, RA-T, and FS-T decreased by 14.3%, 12.1%, 6.4%, and 11.8%, respectively. This showed that coating treatment could effectively reduce water loss from the peduncle during storage, particularly at room temperature.

The PCC of the four varieties was not significantly different at harvest. In the first six days of storage, there were no significant differences between control and coated groups of each variety. During storage coating treatment had different effects on PCC of different varieties. For example, the PCC of RL-T was significantly higher than RL-C from day 12 onwards. FS-T was significantly higher than FS-C from day 18, and RA-T was significantly higher than RA-C on day 18, 30, and 33. On day 30 and 33 the PCC in the coated groups were significantly higher than their controls, indicating that the AA-CaCl$_2$-CMCS-GL coating could delay the degradation of PCC, but to varying extents depending on the cherry variety.

**Table 3.** Mean values for quality parameters: fruit decay ratio, weight loss, and fruit firmness at harvest (0) and after 6, 12, 18, 24, 30, and 33 days storage for control and AA-CaCl$_2$-CMCS-GL coated fruit.

| Characteristic | Storage Time (Days) | RL-C | RL-T | RN-C | RN-T | RA-C | RA-T | FS-C | FS-T |
|---|---|---|---|---|---|---|---|---|---|
| Fruit decay ratio (%) | 0 | 0 ± 0 [a] | 0 ± 0 [a] | 0 ± 0 [a] | 0 ± 0 [a] | 0 ± 0 [a] | 0 ± 0 [a] | 0 ± 0 [a] | 0 ± 0 [a] |
| | 6 | 0 ± 0 [a] | 0 ± 0 [a] | 0 ± 0 [a] | 0 ± 0 [a] | 0 ± 0 [a] | 0 ± 0 [a] | 0 ± 0 [a] | 0 ± 0 [a] |
| | 12 | 1.7 ± 0.6 [a] | 0 ± 0 [b] | 0 ± 0 [b] | 0 ± 0 [b] | 0 ± 0 [b] | 0 ± 0 [b] | 0 ± 0 [b] | 0 ± 0 [b] |
| | 18 | 8.7 ± 1.2 [a] | 1.3 ± 0.6 [b] | 4.3 ± 1.2 [c] | 0 ± 0 [d] | 1.3 ± 0.6 [b] | 0 ± 0 [d] | 0 ± 0 [d] | 0 ± 0 [d] |
| | 24 | 16.3 ± 1.5 [a] | 4.7 ± 0.6 [b] | 10.7 ± 2.5 [c] | 4.3 ± 1.5 [b] | 10.3 ± 2.1 [c] | 1.7 ± 0.6 [d] | 1.7 ± 0.6 [d] | 0 ± 0 [d] |
| | 30 | 35.0 ± 1.0 [a] | 11.3 ± 1.2 [b] | 31.7 ± 1.5 [c] | 8.7 ± 2.1 [bd] | 29.0 ± 4.0 [c] | 2.7 ± 0.6 [e] | 6.7 ± 1.2 [d] | 1.3 ± 0.6 [e] |
| | 33 | 63.0 ± 2.6 [a] | 15.3 ± 0.6 [b] | 58.7 ± 3.1 [a] | 14.3 ± 1.5 [b] | 59.7 ± 5.7 [a] | 5.3 ± 0.6 [c] | 17.7 ± 1.5 [b] | 3.0 ± 1.0 [c] |
| Weight loss (%) | 0 | 0 ± 0 [a] | 0 ± 0 [a] | 0 ± 0 [a] | 0 ± 0 [a] | 0 ± 0 [a] | 0 ± 0 [a] | 0 ± 0 [a] | 0 ± 0 [a] |
| | 6 | 2.23 ± 0.15 [a] | 2.07 ± 0.15 [ab] | 2.04 ± 0.13 [ab] | 1.58 ± 0.26 [c] | 1.81 ± 0.13 [bc] | 1.78 ± 0.19 [bc] | 2.71 ± 0.14 [d] | 1.98 ± 0.31 [ab] |
| | 12 | 4.98 ± 0.24 [a] | 2.88 ± 0.10 [b] | 4.82 ± 0.29 [a] | 2.94 ± 0.06 [b] | 3.80 ± 0.23 [c] | 2.60 ± 0.20 [b] | 4.92 ± 0.20 [a] | 2.83 ± 0.32 [b] |
| | 18 | 5.44 ± 0.26 [a] | 3.47 ± 0.25 [b] | 6.69 ± 0.14 [c] | 4.24 ± 0.14 [d] | 5.25 ± 0.24 [a] | 3.40 ± 0.18 [b] | 7.97 ± 0.21 [d] | 5.19 ± 0.20 [a] |
| | 24 | 6.82 ± 0.27 [a] | 4.85 ± 0.22 [b] | 8.84 ± 0.17 [c] | 5.62 ± 0.13 [d] | 6.47 ± 0.24 [a] | 4.17 ± 0.24 [e] | 9.86 ± 0.28 [f] | 6.42 ± 0.24 [a] |
| | 30 | 9.03 ± 0.25 [a] | 5.87 ± 0.19 [b] | 9.86 ± 0.24 [c] | 7.04 ± 0.27 [d] | 8.01 ± 0.14 [e] | 4.73 ± 0.26 [f] | 11.72 ± 0.23 [g] | 7.45 ± 0.30 [d] |
| | 33 | 11.51 ± 0.30 [a] | 7.39 ± 0.24 [b] | 13.14 ± 0.23 [c] | 8.50 ± 0.23 [d] | 10.75 ± 0.34 [e] | 5.46 ± 0.30 [f] | 14.89 ± 0.33 [g] | 9.44 ± 0.25 [h] |
| Fruit firmness (N) | 0 | 19.49 ± 0.75 [a] | 19.56 ± 0.71 [a] | 19.65 ± 0.71 [a] | 19.81 ± 0.59 [a] | 20.73 ± 0.77 [b] | 21.11 ± 0.66 [b] | 23.87 ± 0.68 [c] | 23.53 ± 0.61 [c] |
| | 6 | 18.95 ± 0.62 [a] | 19.22 ± 0.68 [a] | 19.26 ± 0.64 [a] | 19.51 ± 0.69 [a] | 20.15 ± 0.77 [b] | 20.97 ± 0.69 [c] | 22.68 ± 0.71 [d] | 23.18 ± 0.64 [d] |
| | 12 | 17.91 ± 0.74 [a] | 19.06 ± 0.71 [b] | 18.31 ± 0.69 [a] | 18.97 ± 0.71 [b] | 19.29 ± 0.87 [b] | 20.87 ± 0.53 [c] | 21.55 ± 0.64 [d] | 22.55 ± 0.71 [e] |
| | 18 | 17.04 ± 0.71 [a] | 18.56 ± 0.65 [b] | 17.70 ± 0.65 [c] | 18.59 ± 0.67 [b] | 17.65 ± 0.81 [c] | 20.64 ± 0.58 [d] | 19.83 ± 0.74 [e] | 21.35 ± 0.62 [f] |
| | 24 | 15.27 ± 1.57 [a] | 16.96 ± 0.62 [b] | 17.32 ± 0.61 [b] | 17.75 ± 0.56 [c] | 15.85 ± 0.54 [a] | 20.48 ± 0.87 [d] | 19.10 ± 0.57 [e] | 20.15 ± 0.59 [d] |
| | 30 | 12.23 ± 0.65 [a] | 16.02 ± 0.58 [b] | 16.33 ± 0.69 [b] | 17.18 ± 0.61 [c] | 14.96 ± 0.68 [d] | 20.05 ± 0.81 [e] | 18.61 ± 0.59 [f] | 19.29 ± 0.54 [g] |
| | 33 | 10.39 ± 0.58 [a] | 15.25 ± 0.59 [b] | 13.82 ± 0.59 [c] | 16.49 ± 0.58 [d] | 11.89 ± 0.69 [e] | 19.69 ± 0.53 [f] | 16.22 ± 0.52 [d] | 17.97 ± 0.59 [g] |

Data are mean ± standard deviation of three or ten (firmness) replicate samples. Different letters in the same row indicate significantly different values according to one-way ANOVA followed by Duncan's multiple test range ($p < 0.05$).

**Table 4.** Mean values for skin color characteristics: *L*\*, Chroma, Hue angle at harvest (0) and after 6, 12, 18, 24, 30, and 33 days storage for control and AA-CaCl$_2$-CMCS-GL coated fruit.

| Characteristic | Storage Time (Days) | RL-C | RL-T | RN-C | RN-T | RA-C | RA-T | FS-C | FS-T |
|---|---|---|---|---|---|---|---|---|---|
| *L*\* | 0 | 28.76 ± 2.51 [a] | 30.23 ± 2.61 [b] | 62.95 ± 2.38 [c] | 64.12 ± 2.31 [d] | 32.29 ± 1.25 [e] | 35.64 ± 0.94 [f] | 33.72 ± 2.77 [g] | 35.96 ± 2.58 [f] |
| | 6 | 26.35 ± 2.68 [a] | 28.52 ± 2.34 [b] | 60.81 ± 2.85 [c] | 62.04 ± 2.25 [d] | 31.80 ± 1.07 [e] | 35.05 ± 0.82 [f] | 31.29 ± 2.65 [e] | 33.65 ± 2.36 [g] |
| | 12 | 25.19 ± 2.36 [a] | 27.83 ± 2.17 [b] | 58.23 ± 2.38 [c] | 60.28 ± 2.11 [d] | 31.29 ± 1.18 [e] | 34.65 ± 1.17 [f] | 29.32 ± 2.58 [g] | 31.22 ± 2.18 [e] |
| | 18 | 23.32 ± 1.96 [a] | 26.49 ± 2.28 [b] | 56.53 ± 2.18 [c] | 59.03 ± 2.08 [d] | 30.73 ± 1.14 [e] | 33.97 ± 1.09 [f] | 26.70 ± 2.33 [b] | 30.09 ± 2.11 [e] |
| | 24 | 21.16 ± 1.36 [a] | 24.97 ± 2.33 [b] | 55.03 ± 2.06 [c] | 56.94 ± 2.31 [d] | 28.06 ± 1.04 [e] | 33.34 ± 1.09 [f] | 25.32 ± 2.46 [b] | 29.68 ± 2.23 [g] |
| | 30 | 19.62 ± 1.83 [a] | 22.78 ± 2.42 [b] | 52.42 ± 2.31 [c] | 54.52 ± 2.18 [d] | 26.54 ± 1.03 [e] | 32.98 ± 1.07 [f] | 23.28 ± 2.39 [b] | 28.26 ± 2.15 [g] |
| | 33 | 16.94 ± 1.21 [a] | 20.29 ± 1.96 [b] | 50.03 ± 2.15 [c] | 52.49 ± 2.22 [d] | 22.43 ± 1.05 [e] | 31.85 ± 1.03 [f] | 21.03 ± 2.12 [g] | 26.72 ± 2.33 [h] |
| Chroma | 0 | 20.76 ± 1.56 [a] | 21.05 ± 1.62 [a] | 41.19 ± 1.80 [c] | 41.17 ± 1.77 [c] | 38.56 ± 1.05 [d] | 40.64 ± 0.86 [c] | 34.73 ± 2.88 [b] | 35.99 ± 2.45 [e] |
| | 6 | 21.89 ± 1.75 [a] | 21.44 ± 1.58 [b] | 40.30 ± 1.53 [c] | 40.78 ± 1.61 [c] | 36.53 ± 1.07 [d] | 38.04 ± 1.02 [e] | 35.05 ± 2.39 [a] | 36.08 ± 2.08 [d] |
| | 12 | 25.19 ± 2.36 [a] | 27.83 ± 2.17 [a] | 38.57 ± 1.31 [b] | 40.05 ± 1.66 [c] | 34.86 ± 0.89 [d] | 37.65 ± 0.97 [e] | 32.50 ± 2.41 [f] | 34.81 ± 2.60 [d] |
| | 18 | 20.96 ± 1.59 [a] | 20.52 ± 1.44 [a] | 35.34 ± 1.62 [b] | 38.58 ± 1.31 [c] | 33.79 ± 1.14 [d] | 37.97 ± 1.04 [e] | 30.35 ± 2.36 [f] | 33.44 ± 2.22 [d] |
| | 24 | 19.32 ± 1.78 [a] | 19.57 ± 1.36 [a] | 32.77 ± 1.49 [b] | 36.54 ± 1.43 [c] | 32.86 ± 1.04 [b] | 37.34 ± 1.19 [d] | 29.32 ± 2.17 [f] | 31.90 ± 2.11 [g] |
| | 30 | 17.47 ± 1.29 [a] | 18.38 ± 1.38 [b] | 29.67 ± 1.42 [c] | 35.59 ± 1.20 [d] | 31.75 ± 0.83 [e] | 36.98 ± 0.97 [f] | 27.81 ± 2.31 [g] | 30.92 ± 1.97 [h] |
| | 33 | 13.66 ± 0.30 [a] | 16.99 ± 1.53 [b] | 26.37 ± 1.24 [c] | 32.17 ± 1.34 [d] | 28.45 ± 1.05 [e] | 34.85 ± 1.03 [f] | 26.65 ± 2.18 [c] | 28.92 ± 2.08 [e] |
| Hue angle | 0 | 18.47 ± 2.65 [a] | 19.06 ± 2.99 [a] | 60.73 ± 2.21 [b] | 59.97 ± 2.05 [b] | 25.48 ± 0.89 [c] | 27.37 ± 1.21 [d] | 23.46 ± 3.29 [e] | 24.19 ± 3.01 [e] |
| | 6 | 16.97 ± 2.24 [a] | 18.18 ± 2.59 [b] | 57.98 ± 1.94 [c] | 58.46 ± 2.22 [c] | 23.33 ± 0.63 [d] | 26.37 ± 1.03 [e] | 21.12 ± 3.24 [f] | 21.99 ± 3.03 [g] |
| | 12 | 15.83 ± 2.20 [a] | 16.57 ± 2.55 [a] | 58.48 ± 2.05 [b] | 56.86 ± 1.84 [c] | 22.56 ± 1.12 [d] | 25.93 ± 0.79 [e] | 21.14 ± 3.41 [f] | 21.40 ± 2.75 [f] |
| | 18 | 14.77 ± 1.89 [a] | 16.92 ± 2.32 [b] | 58.95 ± 2.94 [c] | 57.43 ± 2.15 [d] | 20.45 ± 1.03 [e] | 25.21 ± 0.91 [f] | 20.42 ± 2.86 [e] | 20.52 ± 2.64 [e] |
| | 24 | 14.71 ± 1.79 [a] | 16.16 ± 2.21 [b] | 60.09 ± 3.10 [c] | 58.75 ± 1.86 [d] | 18.59 ± 0.89 [e] | 24.32 ± 1.03 [f] | 19.30 ± 3.26 [e] | 18.81 ± 2.12 [e] |
| | 30 | 14.25 ± 1.67 [a] | 16.52 ± 2.22 [b] | 59.11 ± 3.15 [c] | 60.26 ± 2.05 [d] | 18.17 ± 0.93 [e] | 23.74 ± 0.97 [f] | 18.09 ± 3.30 [e] | 18.26 ± 2.05 [e] |
| | 33 | 14.81 ± 2.24 [a] | 15.84 ± 2.13 [b] | 60.15 ± 3.23 [c] | 61.37 ± 2.55 [d] | 16.37 ± 0.86 [b] | 22.49 ± 0.84 [e] | 16.41 ± 2.96 [b] | 17.41 ± 2.02 [f] |

Data are mean ± standard deviation of sixty replicate tests. Different letters in the same row indicate significantly different values according to one-way ANOVA followed by Duncan's multiple test range ($p < 0.05$).

**Table 5.** Mean values for the peduncle freshness parameters: Peduncle Browning Incidence (PBI), Peduncle Moisture Content (PMC), and Peduncle Chlorophyll Content (PCC) at harvest (0) and after 6, 12, 18, 24, 30, and 33 days storage for control and AA-CaCl$_2$-CMCS-GL coated fruit.

| Characteristic | Storage Time (Days) | RL-C | RL-T | RN-C | RN-T | RA-C | RA-T | FS-C | FS-T |
|---|---|---|---|---|---|---|---|---|---|
| PBI (%) | 0 | 0 ± 0 [a] | 0 ± 0 [a] | 0 ± 0 [a] | 0 ± 0 [a] | 0 ± 0 [a] | 0 ± 0 [a] | 0 ± 0 [a] | 0 ± 0 [a] |
| | 6 | 13.7 ± 2.1 [a] | 5.3 ± 1.5 [bc] | 12.3 ± 2.1 [ad] | 0 ± 0 [e] | 10.7 ± 1.5 [d] | 4.3 ± 1.5 [c] | 7.3 ± 1.5 [b] | 0 ± 0 [e] |
| | 12 | 29.0 ± 2.6 [a] | 12.7 ± 2.1 [b] | 24.3 ± 1.5 [c] | 3.7 ± 0.6 [d] | 24.7 ± 2.5 [c] | 9.3 ± 2.5 [b] | 19.0 ± 2.0 [e] | 2.7 ± 0.6 [d] |
| | 18 | 54.3 ± 2.5 [a] | 22.3 ± 3.1 [b] | 44.7 ± 2.1 [c] | 8.0 ± 1.0 [d] | 50.0 ± 2.6 [e] | 15.0 ± 2.0 [f] | 36.7 ± 2.1 [g] | 5.7 ± 1.2 [d] |
| | 24 | 79.7 ± 3.1 [a] | 30.3 ± 2.5 [b] | 57.3 ± 3.1 [c] | 14.3 ± 1.2 [d] | 63.0 ± 3.6 [e] | 19.0 ± 2.6 [f] | 49.3 ± 2.3 [g] | 13.7 ± 2.1 [d] |
| | 30 | 85.0 ± 3.6 [a] | 48.0 ± 3.0 [b] | 74.7 ± 3.8 [c] | 21.3 ± 1.5 [d] | 85.0 ± 3.0 [a] | 25.3 ± 3.1 [d] | 69.3 ± 1.5 [e] | 20.7 ± 1.5 [d] |
| | 33 | 99.3 ± 1.2 [a] | 64.7 ± 2.5 [b] | 88.0 ± 3.0 [c] | 30.7 ± 1.5 [de] | 95.0 ± 2.6 [f] | 33.0 ± 2.6 [e] | 72.0 ± 2.0 [g] | 27.0 ± 2.0 [d] |
| PMC (%) | 0 | 73.2 ± 0.7 [a] | 73.1 ± 0.6 [a] | 76.0 ± 0.4 [b] | 75.8 ± 0.4 [b] | 76.4 ± 1.0 [b] | 76.7 ± 1.5 [b] | 73.6 ± 1.1 [a] | 73.9 ± 0.4 [a] |
| | 6 | 68.3 ± 0.9 [a] | 71.0 ± 0.2 [b] | 70.8 ± 0.6 [b] | 74.9 ± 0.6 [c] | 73.0 ± 0.6 [d] | 76.2 ± 1.8 [c] | 70.6 ± 0.8 [b] | 72.7 ± 0.4 [d] |
| | 12 | 62.8 ± 0.7 [a] | 68.8 ± 0.9 [b] | 67.5 ± 1.3 [c] | 73.3 ± 0.6 [d] | 67.9 ± 0.4 [b] | 76.0 ± 0.7 [e] | 66.3 ± 0.5 [c] | 70.2 ± 0.6 [f] |
| | 18 | 55.9 ± 1.1 [a] | 66.6 ± 0.8 [b] | 65.6 ± 0.9 [b] | 71.1 ± 0.8 [c] | 65.5 ± 0.9 [b] | 75.7 ± 0.6 [d] | 63.4 ± 0.8 [e] | 69.0 ± 0.8 [f] |
| | 24 | 49.7 ± 0.7 [a] | 63.7 ± 1.0 [b] | 62.4 ± 1.1 [b] | 69.3 ± 0.5 [c] | 58.4 ± 1.2 [d] | 74.9 ± 0.5 [e] | 58.6 ± 1.1 [d] | 67.6 ± 0.6 [f] |
| | 30 | 44.5 ± 1.2 [a] | 59.6 ± 0.9 [b] | 58.7 ± 0.8 [b] | 66.7 ± 0.4 [c] | 53.0 ± 1.6 [d] | 74.0 ± 0.7 [e] | 56.5 ± 0.9 [f] | 64.3 ± 0.8 [g] |
| | 33 | 30.5 ± 0.8 [a] | 51.1 ± 1.3 [b] | 43.4 ± 1.0 [c] | 58.6 ± 0.8 [d] | 36.4 ± 1.2 [e] | 69.3 ± 0.8 [f] | 48.5 ± 0.7 [g] | 56.7 ± 0.5 [h] |
| PCC (mg g$^{-1}$) | 0 | 0.464 ± 0.015 [a] | 0.467 ± 0.011 [ab] | 0.475 ± 0.015 [ab] | 0.479 ± 0.011 [ab] | 0.460 ± 0.009 [a] | 0.469 ± 0.007 [ab] | 0.483 ± 0.004 [ab] | 0.490 ± 0.006 [b] |
| | 6 | 0.430 ± 0.016 [ab] | 0.448 ± 0.019 [abc] | 0.440 ± 0.017 [abc] | 0.462 ± 0.009 [c] | 0.425 ± 0.007 [a] | 0.443 ± 0.013 [abc] | 0.437 ± 0.013 [abc] | 0.458 ± 0.008 [bc] |
| | 12 | 0.390 ± 0.012 [a] | 0.424 ± 0.019 [bc] | 0.418 ± 0.011 [bc] | 0.433 ± 0.004 [c] | 0.401 ± 0.005 [ab] | 0.423 ± 0.007 [bc] | 0.414 ± 0.007 [bc] | 0.435 ± 0.007 [c] |
| | 18 | 0.351 ± 0.007 [a] | 0.410 ± 0.014 [bc] | 0.394 ± 0.015 [bcd] | 0.426 ± 0.007 [d] | 0.379 ± 0.011 [e] | 0.401 ± 0.004 [bcd] | 0.393 ± 0.008 [bd] | 0.416 ± 0.008 [ce] |
| | 24 | 0.311 ± 0.006 [a] | 0.363 ± 0.005 [bcd] | 0.356 ± 0.037 [bc] | 0.386 ± 0.006 [cd] | 0.347 ± 0.003 [b] | 0.379 ± 0.006 [bcd] | 0.357 ± 0.011 [bc] | 0.396 ± 0.011 [d] |
| | 30 | 0.264 ± 0.003 [a] | 0.302 ± 0.010 [bc] | 0.314 ± 0.011 [cd] | 0.342 ± 0.003 [e] | 0.295 ± 0.007 [b] | 0.327 ± 0.007 [de] | 0.324 ± 0.007 [d] | 0.363 ± 0.006 [f] |
| | 33 | 0.219 ± 0.003 [a] | 0.277 ± 0.008 [b] | 0.259 ± 0.004 [c] | 0.309 ± 0.006 [d] | 0.247 ± 0.008 [c] | 0.296 ± 0.005 [e] | 0.286 ± 0.004 [be] | 0.335 ± 0.008 [f] |

Data are mean ± standard deviation of three replicate samples. Different letters in the same row indicate significantly different values according to one-way ANOVA followed by Duncan's multiple test range ($p < 0.05$).

As shown in Table 6, PBI had a very significant negative correlation with PMC and PCC, while PMC showed a very significant positive correlation with PCC.

**Table 6.** Correlations between Peduncle Browning Incidence (PBI), Peduncle Moisture Content (PMC), and Peduncle Chlorophyll Content (PCC).

| Characteristic | PBI | PMC | PCC |
|:---:|:---:|:---:|:---:|
| PBI | 1 | - | - |
| PMC | −0.923 ** | 1 | - |
| PCC | −0.901 ** | 0.875 ** | 1 |

Note: ** represent an extremely significant correlation at $p < 0.01$.

### 3.5. SSC, TA, and AA Content

SSC and TA are key factors affecting the eating quality of sweet cherries [17,44] and AA is one of the most important nutrients in many fruit [45]. Changes in these characteristics during storage are shown in Table 7. At harvest, the order of SSC in the cherries was RA-C < FS-C < RL-C < RN-C. TA order was RA-C < RN-C < FS-C < RL-C, while AA order was RN-C < RA-C < RL-C < FS-C.

SSC showed an upwards trend in all groups during storage. On day 12 there was no significant difference in SSC between the coated and control groups of each cultivar. From day 18 to the end of storage SSC in the control groups was significantly higher than in the coated groups, which indicated that the AA-CaCl$_2$-CMCS-GL coating inhibited the rise in SSC. At the end of storage the order of SSC was RA-T < RA-C < FS-T < FS-C < RN-T < RN-C < RL-T < RL-C.

The TA and AA contents in all groups showed a downward trend during storage. From day 6 onwards the TA of all coated cherries (and from day 12 for AA content) was significantly higher than their control group, indicating that the coating treatment could inhibit decomposition of TA and AA.

### 3.6. TPC, TAC, and DPPH RSC

Sweet cherries contain a variety of phenolic substances and high concentrations of anthocyanins which improve antioxidant activity and are beneficial to human health [46]. TPC, TAC, and DPPH RSC during storage of control and treated sweet cherries are shown in Table 8. The varieties exhibited significant differences in all these parameters at harvest and during storage, RL consistently being the highest and RN the lowest.

The phenolics content in all groups showed a downward trend during storage. In the first six days of storage, there was no significant difference in TPC between the control and coated groups. On day 12 there was no significant difference in TPC between the control and coated groups of RL and RA, while the coated groups of RN and FS were significantly higher than their controls. From day 18 to the end of storage TPC of all coated groups was significantly higher than their control group, indicating that coating could sustain TPC during storage. At the end of storage, the order of TPC in the coated groups was RN-T < FS-T < RA-T < RL-T.

TAC showed a downward trend during storage except for RL which increased first then decreased. During storage, there was no significant difference between TAC in RN control and coated groups. However, from day 18 the TAC of the other coated groups was significantly higher than their control groups.

The DPPH RSC of all groups showed a downward trend during storage. On day 6 DPPH RSC of coated groups was higher than their control group, except FS. From day 12 to the end of storage the coated groups were all significantly higher than their control group. At the end of storage, the order of DPPH RSC in the coated groups was FS-T < RN-T < RA-T < RL-T.

**Table 7.** Mean values for the nutritional parameters: Soluble Solids Content (SSC), Titratable Acidity (TA), and Ascorbic Acid (AA) content at harvest (0) and after 6, 12, 18, 24, 30, and 33 days storage for control and AA-CaCl$_2$-CMCS-GL coated fruit.

| Characteristic | Storage Time (Days) | RL-C | RL-T | RN-C | RN-T | RA-C | RA-T | FS-C | FS-T |
|---|---|---|---|---|---|---|---|---|---|
| SSC (%) | 0 | 15.36 ± 0.19 [a] | 15.40 ± 0.08 [a] | 16.62 ± 0.30 [b] | 16.56 ± 0.11 [b] | 11.20 ± 0.36 [c] | 11.40 ± 0.17 [c] | 13.45 ± 0.21 [d] | 13.36 ± 0.19 [d] |
| | 6 | 18.49 ± 0.23 [a] | 17.55 ± 0.19 [b] | 17.49 ± 0.16 [b] | 17.35 ± 0.22 [b] | 12.09 ± 0.23 [c] | 11.55 ± 0.13 [d] | 13.85 ± 0.11 [e] | 13.70 ± 0.12 [e] |
| | 12 | 20.42 ± 0.27 [a] | 20.06 ± 0.51 [a] | 18.86 ± 0.44 [b] | 18.60 ± 0.21 [b] | 12.57 ± 0.26 [c] | 12.05 ± 0.21 [c] | 14.50 ± 0.22 [d] | 14.38 ± 0.16 [d] |
| | 18 | 22.36 ± 0.15 [a] | 21.39 ± 0.10 [b] | 20.81 ± 0.23 [c] | 19.94 ± 0.19 [d] | 13.58 ± 0.27 [e] | 12.55 ± 0.20 [f] | 14.97 ± 0.15 [g] | 14.81 ± 0.14 [g] |
| | 24 | 25.57 ± 0.24 [a] | 24.57 ± 0.24 [b] | 22.73 ± 0.35 [c] | 20.85 ± 0.09 [d] | 14.17 ± 0.14 [e] | 12.99 ± 0.26 [f] | 15.68 ± 0.31 [g] | 15.21 ± 0.18 [h] |
| | 30 | 27.48 ± 0.27 [a] | 25.98 ± 0.26 [b] | 23.88 ± 0.06 [c] | 21.98 ± 0.14 [d] | 14.97 ± 0.15 [e] | 13.75 ± 0.01 [f] | 16.27 ± 0.22 [g] | 15.61 ± 0.22 [h] |
| | 33 | 29.48 ± 0.21 [a] | 27.25 ± 0.38 [b] | 24.67 ± 0.14 [c] | 23.05 ± 0.18 [d] | 15.46 ± 0.41 [e] | 13.83 ± 0.19 [f] | 16.77 ± 0.22 [g] | 16.33 ± 0.20 [h] |
| TA (g·kg$^{-1}$) | 0 | 22.64 ± 0.24 [a] | 22.94 ± 0.27 [a] | 17.75 ± 0.12 [b] | 17.88 ± 0.14 [b] | 15.20 ± 0.14 [c] | 15.00 ± 0.12 [c] | 20.27 ± 0.24 [d] | 20.48 ± 0.21 [d] |
| | 6 | 18.65 ± 0.11 [a] | 20.59 ± 0.12 [b] | 14.15 ± 0.20 [c] | 15.79 ± 0.13 [d] | 12.15 ± 0.28 [e] | 14.64 ± 0.21 [f] | 16.27 ± 0.13 [g] | 19.76 ± 0.17 [h] |
| | 12 | 16.26 ± 0. 18 [a] | 19.31 ± 0.15 [b] | 12.98 ± 0.12 [c] | 14.76 ± 0.12 [d] | 10.49 ± 0.07 [e] | 13.77 ± 0.27 [f] | 14.32 ± 0.21 [g] | 17.12 ± 0.15 [h] |
| | 18 | 15.75 ± 0. 20 [a] | 17.25 ± 0.12 [b] | 9.38 ± 0.22 [c] | 12.56 ± 0.18 [d] | 10.20 ± 0.31 [e] | 12.89 ± 0.21 [d] | 11.17 ± 0.11 [f] | 13.95 ± 0.18 [g] |
| | 24 | 13.70 ± 0.14 [a] | 14.86 ± 0.14 [b] | 8.73 ± 0.13 [c] | 11.31 ± 0.10 [d] | 8.74 ± 0.10 [c] | 12.02 ± 0.17 [e] | 9.79 ± 0.14 [f] | 12.71 ± 0.27 [g] |
| | 30 | 12.40 ± 0.18 [a] | 13.57 ± 0.19 [b] | 8.58 ± 0.14 [c] | 10.01 ± 0.14 [d] | 7.93 ± 0.11 [e] | 11.15 ± 0.20 [f] | 9.03 ± 0.11 [g] | 11.79 ± 0.13 [h] |
| | 33 | 11.00 ± 0.08 [a] | 12.40 ± 0.24 [b] | 8.19 ± 0.15 [c] | 9.66 ± 0.19 [d] | 5.73 ± 0.10 [e] | 9.96 ± 0.27 [d] | 8.28 ± 0.16 [c] | 11.03 ± 0.33 [a] |
| AA (mg·kg$^{-1}$) | 0 | 175.4 ± 1.0 [a] | 175.1 ± 1.3 [a] | 130.0 ± 1.5 [b] | 127.3 ± 1.3 [c] | 142.5 ± 1.1 [d] | 146.5 ± 1.4 [e] | 201.4 ± 1.5 [f] | 198.5 ± 1.7 [g] |
| | 6 | 152.1 ± 1.3 [a] | 152.3 ± 1.8 [a] | 111.4 ± 1.4 [b] | 118.7 ± 1.9 [c] | 125.8 ± 1.3 [d] | 129.9 ± 1.2 [e] | 181.7 ± 1.4 [f] | 184.0 ± 1.4 [f] |
| | 12 | 137.5 ± 1.3 [a] | 144.9 ± 1.8 [b] | 98.6 ± 2.0 [c] | 115.6 ± 1.9 [d] | 118.1 ± 1.2 [d] | 127.2 ± 1.8 [e] | 161.1 ± 1.3 [f] | 175.2 ± 1.3 [g] |
| | 18 | 117.4 ± 1.1 [a] | 132.4 ± 1.5 [b] | 91.8 ± 1.6 [c] | 102.5 ± 1.3 [d] | 120.8 ± 1.1 [d] | 120.8 ± 1.5 [e] | 145.0 ± 1.5 [f] | 164.4 ± 1.3 [g] |
| | 24 | 104.4 ± 1.0 [a] | 124.3 ± 1.4 [b] | 81.6 ± 1.7 [c] | 96.1 ± 1.5 [d] | 96.7 ± 0.9 [d] | 116.6 ± 1.5 [e] | 122.4 ± 1.4 [b] | 153.2 ± 1.2 [f] |
| | 30 | 93.9 ± 1.4 [a] | 112.0 ± 1.1 [b] | 70.8 ± 1.1 [c] | 84.1 ± 1.9 [d] | 80.8 ± 1.7 [e] | 107.9 ± 1.7 [f] | 114.2 ± 1.4 [b] | 144.6 ± 1.6 [g] |
| | 33 | 70.4 ± 1.6 [a] | 93.9 ± 1.5 [b] | 55.0 ± 1.2 [c] | 69.2 ± 1.5 [a] | 62.1 ± 2.0 [d] | 94.4 ± 1.1 [b] | 93.3 ± 1.2 [b] | 118.6 ± 1.8 [e] |

Data are mean ± standard deviation of three (SSC) or four (TA and AA) replicate tests. Different letters in the same row indicate significantly different values according to one-way ANOVA followed by Duncan's multiple test range ($p < 0.05$).

**Table 8.** Mean values for the nutritional parameters: Total Phenolics Content (TPC), Total Anthocyanins Concentration (TAC), and DPPH Radical Scavenging Capacity (RSC) at harvest (0) and after 6, 12, 18, 24, 30, and 33 days storage for control and AA-CaCl$_2$-CMCS-GL coated fruit.

| Characteristic | Storage Time (Days) | RL-C | RL-T | RN-C | RN-T | RA-C | RA-T | FS-C | FS-T |
|---|---|---|---|---|---|---|---|---|---|
| TPC (mg·g$^{-1}$) | 0 | 22.03 ± 0.54 [a] | 22.23 ± 0.30 [a] | 12.20 ± 0.33 [b] | 12.11 ± 0.14 [b] | 20.56 ± 0.46 [c] | 20.55 ± 0.26 [c] | 18.07 ± 0.31 [d] | 18.14 ± 0.21 [d] |
| | 6 | 21.06 ± 0.23 [a] | 21.16 ± 0.41 [a] | 11.69 ± 0.17 [b] | 11.89 ± 0.27 [b] | 19.17 ± 0.55 [c] | 19.90 ± 0.57 [c] | 16.79 ± 0.32 [d] | 17.19 ± 0.36 [d] |
| | 12 | 19.75 ± 0.33 [a] | 20.28 ± 0.38 [a] | 10.39 ± 0.31 [b] | 11.18 ± 0.26 [c] | 17.83 ± 0.52 [d] | 18.56 ± 0.36 [d] | 15.18 ± 0.26 [e] | 16.19 ± 0.28 [f] |
| | 18 | 17.68 ± 0.34 [a] | 19.53 ± 0.40 [b] | 9.14 ± 0.18 [c] | 10.69 ± 0.29 [d] | 16.46 ± 0.48 [e] | 18.38 ± 0.30 [f] | 13.97 ± 0.22 [g] | 15.26 ± 0.29 [h] |
| | 24 | 15.47 ± 0.54 [a] | 19.08 ± 0.28 [b] | 8.41 ± 0.25 [c] | 10.09 ± 0.22 [d] | 14.63 ± 0.51 [e] | 17.48 ± 0.10 [f] | 12.18 ± 0.25 [g] | 14.15 ± 0.24 [e] |
| | 30 | 13.61 ± 0.34 [a] | 18.44 ± 0.21 [b] | 7.20 ± 0.24 [c] | 9.21 ± 0.25 [d] | 13.87 ± 0.48 [a] | 17.00 ± 0.35 [e] | 10.93 ± 0.28 [f] | 12.86 ± 0.22 [g] |
| | 33 | 11.56 ± 0.29 [a] | 17.40 ± 0.22 [b] | 5.74 ± 0.30 [c] | 7.86 ± 0.25 [d] | 11.20 ± 0.35 [a] | 16.59 ± 0.46 [e] | 9.60 ± 0.29 [f] | 11.46 ± 0.29 [a] |
| TAC (mg·g$^{-1}$) | 0 | 1.85 ± 0.02 [a] | 1.84 ± 0.01 [a] | 0.06 ± 0.00 [b] | 0.06 ± 0.01 [b] | 1.05 ± 0.04 [c] | 1.05 ± 0.02 [c] | 0.97 ± 0.02 [d] | 0.97 ± 0.02 [d] |
| | 6 | 2.06 ± 0.02 [a] | 2.06 ± 0.02 [a] | 0.05 ± 0.01 [b] | 0.06 ± 0.01 [b] | 0.95 ± 0.02 [c] | 1.03 ± 0.02 [d] | 0.84 ± 0.01 [e] | 0.87 ± 0.00 [f] |
| | 12 | 1.77 ± 0.02 [a] | 1.82 ± 0.02 [b] | 0.04 ± 0.00 [c] | 0.05 ± 0.00 [c] | 0.76 ± 0.00 [d] | 0.99 ± 0.02 [e] | 0.76 ± 0.01 [d] | 0.79 ± 0.01 [d] |
| | 18 | 1.56 ± 0.01 [a] | 1.70 ± 0.02 [b] | 0.03 ± 0.00 [c] | 0.04 ± 0.01 [c] | 0.70 ± 0.01 [d] | 0.92 ± 0.03 [e] | 0.67 ± 0.02 [d] | 0.71 ± 0.01 [f] |
| | 24 | 1.36 ± 0.02 [a] | 1.50 ± 0.02 [b] | 0.02 ± 0.00 [c] | 0.04 ± 0.00 [c] | 0.65 ± 0.03 [d] | 0.92 ± 0.01 [e] | 0.62 ± 0.01 [d] | 0.70 ± 0.01 [f] |
| | 30 | 1.21 ± 0.02 [a] | 1.34 ± 0.01 [b] | 0.02 ± 0.00 [c] | 0.03 ± 0.00 [c] | 0.57 ± 0.02 [d] | 0.90 ± 0.01 [e] | 0.56 ± 0.01 [d] | 0.62 ± 0.01 [f] |
| | 33 | 1.07 ± 0.02 [a] | 1.26 ± 0.02 [b] | 0.01 ± 0.00 [c] | 0.02 ± 0.00 [c] | 0.47 ± 0.03 [d] | 0.90 ± 0.01 [e] | 0.50 ± 0.01 [f] | 0.54 ± 0.01 [g] |
| DPPH RSC (mg·g$^{-1}$) | 0 | 41.17 ± 0.23 [a] | 41.09 ± 0.18 [a] | 26.06 ± 0.38 [b] | 26.24 ± 0.29 [b] | 39.83 ± 0.73 [c] | 38.86 ± 0.25 [d] | 33.66 ± 0.13 [e] | 33.53 ± 0.25 [e] |
| | 6 | 38.57 ± 0.21 [a] | 39.50 ± 0.28 [b] | 23.36 ± 0.39 [c] | 25.63 ± 0.41 [d] | 34.47 ± 0.48 [e] | 37.52 ± 0.18 [f] | 30.50 ± 0.27 [g] | 31.15 ± 0.40 [g] |
| | 12 | 33.42 ± 0.30 [a] | 38.59 ± 0.14 [b] | 20.20 ± 0.53 [c] | 24.11 ± 0.42 [d] | 29.78 ± 0.48 [e] | 36.32 ± 0.43 [f] | 25.73 ± 0.31 [g] | 28.70 ± 0.35 [h] |
| | 18 | 26.76 ± 0.55 [a] | 37.60 ± 0.35 [b] | 15.66 ± 0.57 [c] | 23.16 ± 0.26 [d] | 23.39 ± 0.60 [d] | 34.26 ± 0.15 [e] | 19.76 ± 0.34 [f] | 24.78 ± 0.36 [g] |
| | 24 | 20.03 ± 0.24 [a] | 36.28 ± 0.24 [b] | 12.32 ± 0.46 [c] | 22.40 ± 0.57 [d] | 19.89 ± 0.32 [e] | 31.54 ± 0.21 [f] | 15.88 ± 0.32 [g] | 22.69 ± 0.41 [h] |
| | 30 | 15.54 ± 0.39 [a] | 35.50 ± 0.33 [b] | 8.61 ± 0.15 [c] | 21.10 ± 0.43 [d] | 15.79 ± 0.55 [a] | 30.31 ± 0.62 [e] | 10.24 ± 0.42 [f] | 19.37 ± 0.42 [g] |
| | 33 | 9.95 ± 0.46 [a] | 34.32 ± 0.38 [b] | 3.48 ± 0.19 [c] | 19.29 ± 0.41 [d] | 6.06 ± 0.32 [e] | 28.38 ± 0.37 [f] | 5.55 ± 0.23 [e] | 14.75 ± 0.50 [g] |

Data are mean ± standard deviation of three replicate samples. Different letters in the same row indicate significantly different values according to one-way ANOVA followed by Duncan's multiple test range ($p < 0.05$).

### 3.7. Correlations between Skin Color Characteristics, TPC, TAC, and DPPH RSC

As shown in Table 9, *L*\* had a very significant positive correlation with chroma and hue angle, and a very significant negative correlation with TPC and TAC. Chroma had a very significant positive correlation with hue angle and a very significant negative correlation with TAC. Hue angle had a very significant negative correlation with TPC and TAC, and a significant negative correlation with DPPH RSC. TPC had a very significant positive correlation with TAC and DPPH RSC. TAC had a very significant positive correlation with DPPH RSC. These results indicate that the skin color of sweet cherry correlated with TPC, TAC, and antioxidant to varying degrees.

**Table 9.** Correlations between skin color characteristics (*L*\*, Chroma, Hue angle), TPC, TAC, and DPPH RSC.

| Characteristic | *L*\* | Chroma | Hue Angle | TPC | TAC | DPPH RSC |
|:---:|:---:|:---:|:---:|:---:|:---:|:---:|
| *L*\* | 1 | - | - | - | - | - |
| Chroma | 0.655 ** | 1 | - | - | - | - |
| Hue angle | 0.972 ** | 0.536 ** | 1 | - | - | - |
| TPC | −0.533 ** | −0.195 | −0.661 ** | 1 | - | - |
| TAC | −0.730 ** | −0.622 ** | −0.775 ** | 0.866 ** | 1 | - |
| DPPH RSC | −0.154 | 0.041 | −0.301 * | 0.893 ** | 0.661 ** | 1 |

Note: ** represent an extremely significant correlation at $p < 0.01$; * represent a significant correlation at $p < 0.05$.

## 4. Discussion

In this study, four different sweet cherry cultivars: one precocious cultivar (RL), two medium maturity cultivars with different skin colors (RN, and RA), and one serotinous cultivar (FS) were chosen. To elucidate the reasons for differences in storability we examined the microstructure of the cherry epidermis and the peduncle surface at harvest. The quality and nutritional characteristics of the four sweet cherry cultivars with and without AA-CaCl$_2$-CMCS-GL coating were measured during storage.

Sweet cherries are susceptible to microbial infection and rot during storage, which affects their quality and leads to economic losses due to shortened shelf-life [2,47]. Factors impacting shelf life include pre-harvest (cultivar, harvest time) and post-harvest factors (temperature and relative humidity of the storage environment, microbiological spoilage, packaging, etc.) [48]. Of these, the cultivar is a critical factor affecting quality after harvest because the genetic make-up determines the structural and chemical compositions of the cultivar [49]. In this study, there were differences in the time and rate of decay of the four cultivars during storage. RL-C began to rot from day 12 of storage and its decay rate was significantly higher than the other varieties. FS-T did not produce any rotten fruit until day 30. By the end of storage, RL-C had the highest decay rate (63.0%) and FS-T the lowest (3.0%). This disparity may be related to characteristics of the different cultivars including surface characteristics, storage durability, and fruit chemistry, etc.

The surface characteristics of the four cultivars were examined. As expected, stomata morphology and wax distribution and morphology showed differences. The skin of RL had small amounts of granular wax and filamentous wax, while the skin of FS was almost completely covered in larger waxy particles. The waxy layer is the interface between the plant surface and the environment. When fungus adheres to and colonizes a plant's surface it contacts the waxy layer first and then spores germinate and produce germination tubes. The completion of these processes often requires a humid environment but water droplets on the wax layer are difficult to aggregate [37]. Thus, wax forms a waterproof layer on the outermost surface which prevents the adhesion, invasion, germination, and reproduction of fungi and pathogenic bacteria. This may explain why FS had the lowest decay rate.

AA-CaCl$_2$-CMCS-GL coating delayed the decay time of all cultivars. The decay rates of the coated groups were significantly lower than their control groups which may be related to the antibacterial properties of CMCS [12] and the addition of calcium to the

coating [40,47]. Calcium inhibits decay by making the cell wall less susceptible to the degrading enzymes produced by pathogens, thus improving cell stability [50].

The weight loss of stored fruit and vegetables is mainly caused by water loss through transpiration and respiration [48,51]. Sweet cherries are an extremely perishable fruit due to their high surface/volume ratio and low resistance to epidermal diffusion [6]. In this study weight loss in all groups showed an upward trend which is consistent with the results of Zam [8], Pettricione et al. [6], and Dong and Wang [52]. Studies have shown that the wax on the surface of fruit and vegetables is closely related to water loss after harvest. Wax is a very effective barrier to water loss, its main physiological function being to prevent uncontrolled water loss via transpiration [37,53,54]. In this study, when stored for 30 and 33 days, the lowest weight loss was seen with RA-C, which may be due to its higher epidermal wax content. FS also had a higher epidermal wax content but due to its higher surface/volume ratio, it suffered greater weight loss during storage.

Fruit firmness is an important indicator of ripeness and storage quality [55]. During the ripening and senescence of sweet cherry firmness gradually decreases [6,30,32]. This is a result of protopectin decomposing to soluble pectin under the action of cell wall degrading enzymes such as pectin galactosidase, polygalacturonase, and pectin methylesterase, which reduce adhesion between cells and the mechanical strength of the cell wall, leading to the softening of fruit tissues and decreased firmness [56]. Studies have shown that edible coatings can effectively retain the firmness of sweet cherries during storage [7,30,32,52,57]. In this study, the firmness of the four cultivars differed at the beginning in line with their various characteristics. During storage, firmness in all groups showed a downward trend, and firmness in the coated group of each variety was significantly higher than its control group, which can be explained by the coating reducing respiration rate and ethylene production, thereby reducing the activity of cell wall degrading enzymes [17].

The skin color of the cherries changed because the concentration of anthocyanins changed during storage [1,58]. This result was consistent with other studies showing that edible coatings delayed skin color change in sweet cherries [6,21,30,57]. The *L*\*, chroma, and hue angle values of the skin of the four cultivars showed significant differences at the beginning of storage, which was related to the concentration of anthocyanins. For example, the *L*\*, chroma, and hue angle of RN-C were significantly higher than the other control groups throughout storage and, correspondingly, the anthocyanin concentration in RN-C was the lowest. The *L*\* values of the coated cherries were significantly higher than the control groups at the beginning of storage, which may be related to the brightening of the surface by the coating. The chroma of RT, RN, and FS all initially increased, which may be related to an increase in anthocyanin synthesis during the post-ripening process after harvest. The subsequent decrease in chroma could be attributed to the decomposition of anthocyanins. By the end of storage, the chroma of all the coated groups was significantly higher than their controls. This could be explained by the oxygen barrier properties of the coating film reducing the oxygen concentration inside the fruit, thereby reducing the activity of phenylalanine ammonia-lyase and flavanone synthase—two key enzymes in the synthesis of anthocyanins [59].

The appearance of the peduncle is a good indicator of the freshness of sweet cherries after harvest [60]. Green and watery peduncles indicate fresh fruit while brown and shriveled peduncles indicate stale fruit [39,61]. Browning and shrinkage are caused by dehydration of the peduncles. Transpiration via the peduncles in an unsaturated atmosphere [62] and osmotic dehydration at 100% relative humidity are the causes of dehydration and atrophy [61]. In this study, the surface microstructure of the peduncle was also examined. There were differences in the number of stomata, wax distribution, and morphology of the peduncle surface of the different cultivars. The surface characteristics of the peduncles may be closely related to PMC during storage. For example, there were more stomata and less wax on the surface of the RN peduncle (which may be related to the peduncle losing more water during storage), while there were no stomata and more wax on the surface of the FS peduncle (which may be related to lower water loss). Tissue browning of fruit

and vegetables during postharvest storage is caused by the formation of quinones from phenolic substances under the action of polyphenol oxidase (PPO) [63]. PBI is, thus, closely related to PPO activity which is also related to oxygen content. This explains why PBI in the coated groups was significantly lower than their control groups—the oxygen barrier properties of the coating limited oxygen entry to the interior of the peduncle and reduced the activity of PPO, suppressing PBI. The PCC also showed a downward trend during storage but PCC in the coated groups was significantly higher than their control groups at the end of storage, which may be related to chlorophyll decomposition and the oxygen barrier property of the coating.

SSC and TA determine sweetness and acidity which are important parameters determining the quality, taste, and consumer acceptance of sweet cherries [1,17]. AA is one of the most important antioxidant nutrients in fruit and plays a variety of biologically active functions in the human body including the scavenging of free radicals produced in the fruit [45,64]. Starch is hydrolyzed to monosaccharides through respiration and other catabolic processes resulting in an increase in SSC during storage [30]. Loss of water also increases SSC but the decomposition of starch plays a major role [17]. Other reports, consistent with this study, have shown that edible coatings can delay the rise in SSC by inhibiting respiration [6,8,43]. Since organic acids are usually the main substrate for respiration and other metabolic processes, the TA content exhibited a downward trend during postharvest storage [30]. This study observed that TA and AA contents of sweet cherries in all groups showed a downward trend and that coating treatment inhibited the decomposition of TA and AA due to the oxygen barrier effect of the coating reducing oxygen concentration inside the fruit, reducing the intensity of respiration and related enzyme activities, thereby reducing the consumption of TA and the decomposition of AA [65].

Genetic, pre-harvest, post-harvest, and other factors affect the nutrient content of sweet cherries. These nutrients have a variety of biological activities with antioxidant, anticancer, and anti-inflammatory effects. Phenolics and anthocyanins are found in the peel of sweet cherries which improves their sensory quality (color) and taste (astringency) [66]. Studies have shown that the TPC in cherry peel is four to five times higher than in the pulp [43]. These compounds also have a strong antioxidant capacity and can capture free radicals generated during oxidative stress [67]. In this study, the TPC, TAC, and DPPH RSC of the cultivars at the beginning of storage differed significantly, as governed by their genetic make-up. These parameters remained highest in RL during storage while RN had the lowest. This also relates to skin color since $L*$ was shown to have a very significant negative correlation with TPC and TAC. The edible coating treatment helped to maintain TPC and TAC (except in RN), probably due to the oxygen barrier properties of the film. The TPC and TAC of RN were significantly lower than the other varieties which may be related to its yellow-red peel color. DPPH RSC correlated positively with TPC and TAC, which was consistent with the results of Carolina Formica-Oliveira et al. [68] and Nair et al. [67].

## 5. Conclusions

There were significant differences in the surface microstructure of four sweet cherry epidermis and peduncle at harvest which was closely related to fruit decay ratio during storage. The AA-CaCl$_2$-CMCS-GL coating treatment delayed decay and the fruit decay ratio of the coated groups were significantly lower than their controls. The edible coating treatment also helped to maintain quality and nutritional characteristics of four sweet cherry cultivars during storage, including reducing weight loss of the fruit, enabling the sweet cherries to maintain better skin and peduncle color, and higher fruit hardness, TA, AA, TPC, TAC, and DPPH RSC. These results suggested that AA-CaCl$_2$-CMCS-GL coating could be considered as a preservation method for improving postharvest quality and nutritional properties of different sweet cherry cultivars, and when using a coating to preserve sweet cherries, the cultivar should be considered, and an appropriate cultivar should be selected to achieve the best preservation effect.

**Supplementary Materials:** The following are available online at www.mdpi.com/2079-6412/11/4/396/s1, Figure S1: the pictures of four cultivars of sweet cherry at harvest. Figure S2: standard curve for the determination of TPC, Figure S3: standard curve for the determination of DPPH RSC.

**Author Contributions:** Conceptualization, Q.-L.C. and Y.W.; methodology, Q.-L.C., Y.-L.Z., F.S., H.F., Y.-Q.Z., S.-T.L., Z.-H.L., L.L, and Y.-K.S.; formal analysis, Y.-L.Z. and Q.-L.C.; resources, Y.-L.Z., H.F., Y.-Q.Z., S.-T.L., Z.-H.L., L.L. and Y.-K.S.; writing—original draft preparation, Y.-L.Z.; writing—review and editing, Y.-L.Z., Q.-L.C. and F.S.; supervision, Q.-L.C. and Y.W.; project administration, Y.W.; funding acquisition, Y.W. All authors have read and agreed to the published version of the manuscript.

**Funding:** This research was funded by the Shanxi Provincial Key Research and Development Project (Agricultural sector) (CN), No. 201903D211007-1.

**Institutional Review Board Statement:** Not applicable.

**Informed Consent Statement:** Not applicable.

**Data Availability Statement:** No new data were created or analyzed in this study. Data sharing is not applicable to this article.

**Conflicts of Interest:** The authors declare no conflict of interest.

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
