# Peer review of "Effect of Edible Carboxymethyl Chitosan-Gelatin Based Coating on the Quality and Nutritional Properties of Different Sweet Cherry Cultivars during Postharvest Storage"

_coatings, doi:10.3390/coatings11040396_

Round 1

Reviewer 1 Report

The Authors have presented a significant number of analysed materials and results. The submitted work is very impressive. The performed analyses are well designed and presented.  In my opinion, after minor revision, the work can be published. Below I indicate some discrepancies which need  to be taken into consideration:

In the introduction, the role of AA and CaCl2 for CS as well as CMCS-GL has to be explained. Moreover, using both CaCl2 and AA together has to be justified.

Line 127 – “The solutions were mixed in a ratio of 2:1” It has to be clearly indicated if it was mass or volume ratio. Moreover, there is no information why the 2:1 ratio was analysed.

Line 128 – “then 2 % CaCl2 powder, and 2 % L-AA were added to the solution” – 2% in relation to which of the compounds?

The use of the surfactant (tween-20) has to be explained.

The Authors indicate that during centrifugation the particulates have been removed. Information needs to be included relating to the composition of the particulates.

The abbreviations such as PBI, PMC, and PCC have to be explained.

There is no information concerning the Glutaraldehyde in the “Chemicals” section. An explanation why glutaraldehyde, which is known as a toxic component, was used in this study and why the skin of cherries was immersed in the glutaraldehyde solution.

Reviewer 2 Report

The authors evaluated the topic Effect of edible carboxymethyl chitosan-gelatin based coating on the quality and nutritional properties of different sweet cherry cultivars during postharvest storage.

The article is well written and I recommend publishing it after minor corrections.

Figures 1 and 2 should be placed after the text where they first appear.

Reviewer 4 Report

The manuscript deals with the effect of edible carboxymethyl chitosan-gelatin based coating on the quality and nutritional properties of different sweet cherry cultivars during postharvest storage.

Abstract

This section is vague. Please present your main results (obtained values).

Introduction

The topics must be better linked.

Materials and methods

Line 124- “2.2.1. Preparation of AA-CaCl2-CMCS-GL edible coating solution”??characterization of the solution??viscosity??wettability??contact angle??color??opacity??

Line 134- “The four cultivars of sweet cherries were washed in tap water, dried at 23 ± 1 ℃, immersed in AA-CaCl2-CMCS-GL edible coating solution for 2 min, then dried completely at 23 ± 1 ℃. The control group was washed with tap water.”???amount of coating solution used per amount of sample??thickness of coating solution after drying??

Line 179- “A colorimeter (model CM–5; Konica-Minolta, Japan) was used to measure the skin color (L*, a*, and b*) of sweet cherry on opposite sides of 30 fruit. L* chroma, and hue angle values were used to describe the skin color characteristics using the following calculations [17,32,43]:”??illuminant used??ºobserver???

Results

Pictures of each sample??

Conclusion

Please do not repeat your results and focus on your main conclusions.

Round 2

Reviewer 4 Report

Materials and methods

 “A colorimeter (model CM–5; Konica-Minolta, Japan) was used to measure the skin color (L*, a*, and b*) of sweet cherry on opposite sides of 30 fruit. L* chroma, and hue angle values were used to describe the skin color characteristics using the following calculations [17,32,43]:”??illuminant used, D65??ºobserver???2º or 10º??
